# An Investigation of the Implementation of Obligatory Physical Activity Classes for 5th–7th Grade in Norway

**DOI:** 10.3390/ijerph192114312

**Published:** 2022-11-02

**Authors:** Erling Algroy, Oddrun Samdal, Ellen Haug

**Affiliations:** 1Department of Teacher Education, NLA University College, Bergen, Pb 74 Sandviken, 5812 Bergen, Norway; 2Department of Health Promotion and Development, University of Bergen, 5020 Bergen, Norway

**Keywords:** physical activity, guidelines, school commitment, PE teachers

## Abstract

In 2009, all Norwegian 5th–7th graders were allocated 76 h of obligatory physical activity (PA) classes in addition to physical education (PE). The study explores how schools implemented these classes and the relationship with school physical activity guidelines. The sample consisted of 134 schools participating in the WHO collaborative Health Behaviour in School-Aged Children (HBSC) survey in 2014 (*n* = 69) and 2018 (*n* = 65). Ten questions concerning PA were asked in 2014, and four of these were included in the 2018 survey. Chi-squared tests were used to investigate differences between groups. In 2014, 51% reported that PE teachers led the classes; this had reduced to 30% in 2018. A combination of teacher- and student-driven activities was most common. More student-led activities were observed when nonpedagogical personnel were responsible for the classes. Most schools reported no professional staff development related to the implementation of obligatory PA. In 2018, schools with written guidelines on physical activity had to a greater extent implemented staff development measures and increased the use of PE teachers compared to 2014. A considerable variation regarding teaching competence, teaching forms, group sizes, and facilities makes the outcome of the PA scheme uncertain. A potential effect of having established written school policies on the implementation of physical activity classes was however found.

## 1. Introduction

On 1 August 2009, all pupils in years 5–7 in Norwegian schools were allocated compulsory physical activity (PA) lessons in addition to physical education (PE) lessons. The idea behind this national regulation of obligatory PA was to facilitate a more varied school day for pupils [1]. This decision came on the back of research demonstrating that a large percentage of young people in Norway did not meet the recommended levels of daily physical activity and that the number of overweight people in this age category was substantial [2]. Recent research shows that the percentage of young people in Norway meeting the recommended daily levels of physical activity drops significantly between the ages of 9 and 15: from around 90% to around 70% [3]. This coincides with international research that confirms that the time young people spend on physical activity appears to decline with time and advancing age. Meanwhile, the time spent on stationary screen-based activities increases [4]. This development can be considered a cause for concern, as physical activity is connected to several health benefits, as well as prevention of lifestyle illnesses for children and young people [5].

The Norwegian government focused on physical activity with its Action Plan on Physical Activity 2005–2009 [6]. This represented a mobilization for better health through increased physical activity. One of the goals was at least an hour of physical activity for children and young people. Pursuant to Section 2 and Section 3 of the Education Act, the following regulations were adopted on 1 August 2009 for physical activity in primary education.

Pupils in years 5 to 7 must always have physical activity lessons outside of physical education classes. The physical activity lessons will amount to 76 h within the period of years 5 to 7, c.f. the distribution of lessons and hours. The physical activity should be arranged so that all pupils, regardless of ability level, can experience joy, mastery, togetherness, and variation in the school day. The rules regarding individual assessment in chapter 3 and the rules on the requirement for pedagogical competence for teaching staff in chapter 14 do not apply. Otherwise, the regulations associated with the Education Act apply [7].

The political initiative to achieve better health among the population through physical activity is in line with national [8] and international guidelines [9]. The integration of physical activity to a larger extent into key settings of children’s lives is an important step towards increasing activity levels and establishing good activity habits, and school is defined as a key area for the implementation of various health-promoting measures [8,9]. The introduction of the obligatory PA beyond PE is therefore an example of national policy setting guidelines at a school level, which in turn can affect individual pupils’ activity levels. This approach represents a socioecological perspective, which highlights that the individual health behavior is complex and is affected by a number of factors at different levels and the interaction between these factors [10]. McLeroy et al. [11] outline five different levels, with state policy as the starting point, which in turn affects the societal level, the institutional level, the interpersonal level, and finally the individual level [11]. Thus, this sociocultural ecological perspective expands from the reciprocal determinism theory [12] and addresses how a person’s behavior is influenced by both the personal factors and the social environment in a bidirectional manner to include multidimensional connections among social and cultural factors within one’s environment [11]. As physical activity is a multifaceted behavior, the usefulness of a socioecological approach has been widely accepted and emphasized [13].

As stated in the purpose of the scheme, the Directorate for Education and Training emphasizes the importance of physical activity and its benefits for the learning environment, learning outcomes, and physical and mental health [1]. These beneficial effects have been supported by an increased amount of research in the field [5,14]. The scheme is considered to be educational in the same way as school subjects according to the Education Act, and pupils therefore have the right to adapted training. The content of the PA sessions must therefore be customized so that all students can achieve a satisfactory result, which corresponds to the purpose of the activities, regardless of their ability or any other preconditions. The rules on special education also apply to PA. However, the Norwegian Directorate for Education and Training (2009) points out that the PA scheme is not part of the formal curriculum and there are therefore no specific competence aims set within the scheme. For that reason, there is no requirement for school leaders to specifically use staff with pedagogical qualifications or expertise in physical activity for the sessions. The school leader must decide on the level of competence required to lead the session, but they should take into account the pupils’ interests and abilities when designing the activity plan [1].

The introduction of the obligatory PA scheme addresses the institutional level and involved an extension of the school day for the year groups concerned, with clear guidelines on how the time can be managed. The scheme corresponds in practice to 40 min of physical activity per school week per year. There is no opportunity to use, for example, break times for this. Although the schools have a great deal of freedom to allocate the 76 extra hours, they do not have the opportunity to accumulate the time to use it on sports days, skiing days or the like. Schools are also not allowed to divide the hours into such small chunks that they are not fit for purpose [1].

Following the introduction of PA, seven core requirements for the scheme were established for the institutional school level and were to be implemented by the school leaders. These minimum requirements were communicated in a circular from the Norwegian Directorate for Education and Training (2009) and schools are required to ensure that they meet them. The seven minimum requirements are as follows:The activity must promote physical and mental healthThe activity should give pupils joy and a sense of masteryThe activity should contribute to improving motor skillsIt should be arranged so that varying activities make for varied school daysThe activities should occur regularlyAll pupils must be given the opportunity to participate, regardless of functional ability or other preconditionsThe activities should promote social competence

A key feature of the initiative was to distinguish the PA lessons from PE by not setting competence objectives and not requiring the pupils to wear sport clothes for these classes. The requirements in combination with the compulsory participation intend to stimulate both the interpersonal level and the individual level in the socioecological model [11]. The interpersonal level can be observed in that the class does physical activities together, stimulating togetherness and joint activity. At individual level, the initiative intends to facilitate experiences of having fun while being physically active and providing a framework of routine to be active that can stimulate the level of physical activity also outside of school hours.

In order to have the chance to find out if the investment in PA had the desired effect, and to further learn about best practice, an advisory group for the obligatory PA scheme stressed the importance of evaluation of the project [15]. So far, one report has been published, which summarizes various studies on the introduction of PA in schools. In the report, eight different projects were included, which looked more closely at how the introduction of PA in the period 2009–2013 worked in schools [16]. Key findings from the report included the extensive use of staff without formal pedagogical competence in the PA sessions, significant variation within the student groups when it came to activity level, and challenges in getting inactive students to participate in the sessions. There was also little variation in content, and year 5 displayed a higher level of activity than year 7. However, it did appear that the PA sessions were indeed scheduled. The report points out that there is some uncertainty regarding the methodological quality of the aforementioned projects, and it is therefore difficult to say with certainty what works well and what works less so with the introduction of PA to years 5–7. The projects referred to in the report indicated that schools had difficulty in meeting the minimum requirements set for the scheme, primarily due to the use of non-teaching staff. The report concludes that there is a need for a thorough evaluation of the scheme.

It has now been 13 years since the focus on PA was introduced to schools. Considerable resources are used each year to achieve the goal of a more active daily life for children in years 5 to 7. So far, research that examines the implementation of the obligatory PA scheme has been limited, and we therefore know little about whether schools engage with the scheme in line with the guidelines and recommendations that are provided to them. The purpose of this study is to analyze how a larger sample of Norwegian schools manage the sessions and examine whether there is a connection between the school’s commitment to physical activity through written guidelines and organization of the PA sessions.

## 2. Materials and Methods

### 2.1. Sample

The data were collected as part of the Norwegian part of the international study Health Behaviour in School-Aged Children: A WHO collaborative Cross-National Survey (HBSC), which collects nationally representative data every four years on 11-, 13- and 15-year-olds through a school-based survey. The survey also collected data from the head teachers of the selected schools, who were asked to answer a questionnaire to map the school’s conditions for the development of positive health habits and a good psychosocial environment, including the use of the 76 school hours allocated to the PA initiative. The study is based on school-level data from the survey years 2014 and 2018. The Department for Health Promotion and Development (HEMIL) at the University of Bergen was responsible for collecting the data.

### 2.2. Sampling Procedure

School classes and schools were selected from a geographically stratified list to ensure a nationally representative sample. Questionnaires for head teachers were sent to 138 primary schools in 2014 and 136 primary schools in 2018, with 69 schools answering the survey in 2014 and 65 schools in 2018. The total number of responses was therefore from 134 different schools, with a response rate of 49%.

### 2.3. Questionnaire

The questionnaire for school leaders was developed in a cross-national collaboration for the HBSC study, and piloted among Norwegian school principals or their deputies to test its face validity [17], and was supplemented with national questions for the Norwegian setting about the PA scheme. In the questionnaire that was sent to the schools in 2014, 10 of the questions dealt with the PA scheme at years 5–7, and four of these questions were repeated in the 2018 survey. Questions relating to written guidelines on physical activity and skill development were asked in both the 2014 and 2018 surveys.

### 2.4. Distribution of the Sessions

The following question was asked on the use of the allocated 76 h PA: “How are the sessions distributed across the year groups?” There were five alternatives: (1) one hour in year 5 and one hour in year 6, (2) one hour in year 6 and one hour in year 7, (3) one hour in year 5 and one hour in year 7, (4) 40 min during each year, and (5) other (please specify). Furthermore, the following question was asked: “How is the physical activity session distributed throughout the week?”, with two answer options: (1) carried out as one continuous session, and (2) carried out as two or more sessions per week. The head teachers were also asked about time points with the question “When during the day are the physical activity sessions mainly carried out?”, with four answer options: (1) at the beginning of the day, (2) in the middle of the day, connected to the break, (3) at the end of the day, and (4) it varies (specify). This question was asked in both the 2014 and 2018 surveys.

### 2.5. Organization of Sessions

With regard to who is responsible for the PA sessions, the following question was asked: “Who is responsible for conducting the physical activity sessions?” This question was posed in both 2014 and 2018. There were three alternatives: (1) pedagogical teaching staff who teach PE, (2) pedagogical teaching staff who do not teach PE, and (3) other staff or external partners (please specify).

In order to survey the size of the classes, the question was asked (2014): “How large is the group of pupils who take part in physical activity together?”, with three alternatives: (1) under 25 pupils, (2) 25–50 pupils, and (3) over 50 pupils. Furthermore, the head teachers were asked (2014 and 2018): “How are the sessions mainly organized?”, with four alternatives: (1) teacher-directed activities, (2) pupil-directed/self-governed, (3) combination of teacher-directed and pupil-directed, and (4) other (please specify).

The head teachers were also asked in 2014 which physical areas were used for the PA sessions. The areas that were listed were sports halls, other indoor areas, outdoor courts/pitches, other outdoor areas, outdoor areas away from school premises, and indoor areas away from school premises, with all the options on of a Likert scale. For each area, there were the following five response options: (1) to a very small extent, (2) to a small extent, (3) to some extent, (4) to a large extent, and (5) to a very large extent.

### 2.6. Competence-Enhancing Measures

The following question was asked in both 2014 and 2018: “Have any form of competence-enhancing measures been carried out among staff in connection with the introduction of the two hours of physical activity?” The options were (1) no and (2) yes (specify).

### 2.7. Physical Activity Policies and Skill Development

To assess whether a school had a physical activity policy, the head teachers were asked in both in the 2014 and 2018 surveys whether the school had guidelines for creating a stimulating school environment and new opportunities to be physically active during breaks, during school hours and after school hours. The response alternatives were: (1) yes—written guidelines, (2) yes—informal guidelines (verbal), and (3) no. They were also asked in 2014 and 2018 whether the school had plans for staff (as part of their professional development) to take further education courses in sports and physical activity. The response alternatives were: (1) yes, (2) no, and (3) no, at least one member of staff already has these qualifications.

The final two questions were slightly more open in that there were no answer categories: “Which resources are used in the conduction of the physical activity sessions?” and “Is the school satisfied with the scheme? (Please elaborate).” These two questions were asked only in the 2014 survey.

### 2.8. Analysis

The data were analyzed in SPSS version 27. Statistical differences were investigated using chi-squared tests with a statistical significance level of *p* < 0.05. When significant differences were detected, effect size (ES) was calculated with Cramer’s V. An ES > 0.07 was considered small, >0.21 medium, and >0.35 large [18,19].

## 3. Results

Around half (49%) the schools (2014) stated that the group sizes in the PA sessions were under 25 pupils. Also, many schools (39%) stated that the size of the group of pupils was between 25 and 50, and 12% stated that the groups for the PA sessions contained over 50 pupils. The size of the pupil groups in the PA sessions is presented in Figure 1.

A majority of schools (68%) conducted the PA sessions as one continuous session, while 32% stated that the sessions were broken up into two or more sessions per week.

The most popular time distribution was to allocate 40 min weekly to each year group. Of the head teachers questioned, 45% said they used this time allocation at their school, while a total of 25% stated that they chose to allocate one hour weekly across two years, and 31% stated that they had different ways of distributing the time (Figure 2).

In terms of which indoor areas were utilized to a large or very large extent in 2014, 59% reported using sports halls and 14% other indoor areas. Of the outdoor areas that were reported to be used to a large or very large extent, 63% reported using outdoor courts/pitches and 55% reported using other outdoor areas. When it came to the use of outdoor areas away from school premises, the answers were more varied, with 19% of head teachers stating that the school utilized such areas to a large or very large extent. Indoor premises away from the school were used to a small or very small extent by a majority (89%). When asked if the schools were satisfied with the scheme, around two-thirds of the head teachers answered that they were satisfied.

### Comparison of Results from 2014 and 2018

Around half (51%) the schools reported in 2014 that pedagogical staff who teach PE were responsible for conducting the PA sessions. In 2018, this proportion dropped considerably to 31%. Thirty percent of the schools stated in 2014 that it was “other staff” who were responsible for the PA sessions, and this proportion increased to fifty percent in 2018. When the schools specified who the “other staff” were, it was mainly teaching assistants that were used. The number of pedagogical staff without subject-specific expertise was unchanged between the two years. An overview of the type of staff who were responsible for the PA sessions is provided in Table 1.

The difference was significant in the distribution for 2014 and 2018, with a moderate effect size (*p* = 0.039, ES = 0.24). The main difference between the two survey years is that fewer schools reported sessions being teacher-led in 2018 (24%) than in 2014 (40%), and that there appears to have been a shift towards self-governed by pupils’ autonomy with an increase in schools combining pupil-led/self-governed activities and teacher-led activities from 49% in 2014 to 60% in 2018. The proportion who stated that sessions were mostly pupil-led/self-governed activity was 12% in both 2014 and 2018.

If we look at the connection between who was responsible for the undertaking of the sessions and which form of organization that was mainly used, there were large and significant differences in 2014 (*p* = 0.001, ES = 0.38) between the groups. Pedagogical staff who taught PE chose teacher-led activities to a greater extent, and generally did not allow self-governed activities. When teachers who taught PE were responsible for the sessions, approximately twice as much teacher-led activity was reported in 2014 than in 2018 (62% versus 33%). Both PE teachers and teachers without PE education seemed to avoid self-governed activities. This way of organizing the sessions was mostly favored by “other staff,” where over 25% of the schools in both 2014 and 2018 reported that this was their most common means of organizing the PA sessions. An overview of the kind of staff and the means of organizing the sessions is demonstrated in Figure 3.

A majority of schools stated in both 2014 and 2018 that it varied when in the day the PA sessions were conducted. There were no major changes between the two years. An overview of when in the day the sessions were conducted can be found in Figure 4.

In 2014, 16% of schools reported that they had introduced competence-enhancing measures for their staff in connection with the introduction of obligatory PA. In 2018, this proportion had increased to 34%. Various courses were named among these competence-enhancing measures, for example, activity courses run by local municipalities and different leadership courses.

In the 2018 survey, we found that written plans for PA-competence development correlated positively with the use of pedagogical staff in the PA sessions (*p* = 0.001, ES = 0.46). There was also a correlation (*p* = 0.046, ES = 0.32) between schools that had written guidelines for the physical activity and those that had conducted competence development related to PA, and between schools that had made plans to further educate their staff in the field of physical activity and those that had undertaken competence development related to PA (*p* = 0.045, ES = 0.32). We found no associations between written guidelines and the PA scheme in the 2014 survey.

## 4. Discussion

The purpose of this study was to investigate how Norwegian schools have managed the implementation of 76 h obligatory PA between grades 5 and 7. The main findings are that the schools used relatively few professional teachers for these sessions and that the proportion decreased from 2014 to 2018. The use of teaching assistants, on the other hand, increased, and half the schools stated in 2018 that it was this type of staff who were predominantly responsible for the PA classes. At schools where this was the case, we found that 25% (2018) and 30% (2014) of these schools organized the sessions mainly as pupil-led/self-governed. Correspondingly, the proportion was 0 (2014) and 9% (2018) at schools that used PE teachers for the sessions. In general, however, the most common means of organizing the sessions was a combination of pupil-led/self-governed and teacher-led activities. We also observed an increased use of this approach in 2018 compared to 2014, and a corresponding decrease in teacher-led organization. A higher proportion of schools in 2018 had carried out skills training for staff compared to 2014. In 2018, we also found a correlation between having written guidelines and plans for investing in physical activity and having conducted skills training for the staff related to the PA scheme.

Before the introduction of the scheme, the national advisory group for the scheme recommended that as many as possible of those who were to work with the PA sessions should be teachers and have at least 30 study points in PE in their educational background [15]. The findings that few professional teachers and even fewer PE teachers were leading the PA sessions is worrying. Studies show that teachers with a background in PE are able to teach the subject more effectively [20,21,22] and are able to achieve a higher level of engagement and activity than those without this subject-specific educational background [23,24,25]. It is thus reasonable to assume that a similar relationship applies to teacher-led PA sessions. One of the requirements for the PA sessions is that there must be a health-promoting element, so it is important to ensure a sufficient level of activity in order for pupils to achieve the goals related to physical health. Sallis et al. [26] conducted an intervention study over the course of two years where they evaluated the effects on health and compared the results obtained from teachers with formal PE competence and those without this subject-specific background. Pupils whose PE lessons were led by a teacher with formal PE competence displayed a higher level of activity during the sessions and notably greater health benefits in the form of both increased strength and increased endurance after the two-year period [26]. Our results showed that in 2018, only 30% of the schools used staff with formal PE competence for the PA scheme and that half the schools reported that they used staff without any form of pedagogical competence for these sessions. Our findings showed a significant use of staff without specific pedagogical competence and an extensive use of classroom assistants and special educators primarily responsible for the PA sessions. These findings were consistent with the results from a 2011 project wherein 153 schools in Bergen were asked about the PA scheme [16]. Formal PE competence is not, however, a requirement for the PA sessions, and the use of staff without this subject-specific competence is therefore not contrary to the guidelines. At the same time, the extensive use of staff without PE-specific competence is cause for concern in terms of ensuring good quality and fulfilling the objectives of the scheme. To ensure that the students experience joy and a sense of mastery, that their interests and abilities are taken into account, and that everybody can participate regardless of ability or condition, PE-specific competence would nevertheless be of great importance [22,27].

Our findings do not indicate that competence-enhancing measures are prioritized in order to address the lack of formal competence in the physical activity subject area, as 84% of schools reported in 2014 that no form of competence training had been conducted since the introduction of the PA scheme. In 2018, however, there was an increased proportion who had carried out competence training in PA, but there was still a majority (66%) who had not. Measures to raise the competence of those responsible for physical activity are important [15,26,28,29]. We also found in our results that there is a connection between formal competence in physical activity and how the sessions are generally organized. This is evident, among other aspects, from the fact that the category “pupil-led/self-governed activity,” whereby pupils mainly control the content and organization of the sessions, was used to a very small extent when pedagogical staff had the responsibility for the sessions. However, this means of organizing the sessions was not uncommon when teaching assistants had responsibility for the sessions. Despite the fact that research indicates that pupils are on average more active while participating in activities during break times compared to teacher-led PE lessons, the differences in activity level are greater between groups that are motivated and those that are not motivated during activity sessions [30]. Previous studies have shown teachers’ impact on pupils’ motivation in PE [31], and also found higher activity levels among motivated pupils [30]. Measures to raise the competence levels of those responsible for the PA sessions thus become more important in respect to the quality and purpose of the sessions.

Similarly, the report “Mapping research and evaluation: the introduction of 76 h of physical activity in years 5–7” points out that there have been challenges in getting inactive students to participate in the PA sessions, that there is little variation in the activities offered during the PA sessions, and that this is connected to a lack of competence among those responsible for the PA scheme at the schools [16]. This report also points out that there is reason to believe that not all pupils participate, or are given the opportunity to participate, with those in most need of the PA sessions often not getting the opportunity to participate. Pupil participation is highlighted as a fundamental principle of the scheme. Autonomy is important in terms of pupils’ motivation for physical activity in PE [32], and there is reason to believe the same applies to PA. At the same time as pupil participation is encouraged, schools also have a responsibility to ensure the quality of the sessions. The fact that schools reported the sessions often taking the form of pupil-led/free activities/self-governed does not necessarily indicate a high degree of autonomy for all pupils. It is thus very important that the PA sessions are not simply left to the pupils, but that there are staff present who ensure the quality of the sessions, and—preferably with the pupils—develop these sessions to create an environment of enjoyment and good experiences with physical activities for all participants.

Our findings from 2014 showed that relatively large pupil groups for the PA sessions were common. For 12% of the schools, the groups consisted of over 50 pupils, while 39% reported having groups of 25–50 pupils. Studies show that large groups in PE classes lead to less activity among pupils [33], and the same most likely applies to PA sessions. If we look at the extensive use of staff without PE-specific competence in relation to the group sizes for the PA sessions, there would be reason to question whether some schools are indeed fulfilling the aims of the PA scheme, such as contributing to improving motor skills, promoting physical and mental well-being, and providing joy and a sense of mastery [1]. The areas in which the PA sessions are carried out will also be important, especially when it comes to large groups of pupils. Information on the size of the groups was not asked for in the 2018 survey, but many of the same patterns were repeated between 2014 and 2018, and there is little reason to believe that group sizes would be any exception.

In addition to the reported results, several head teachers answered in the open question sections that school subjects are generally prioritized over the PA scheme. Other findings that support this include schools reporting a large degree of variation in relation to the distribution of sessions between the year groups and the time of day they were undertaken. Several schools reported that it varied when the different year groups had their PA sessions and that they did not have fixed days or times for the sessions. This was also the case in terms of facilities, with several schools reporting challenges with access to both equipment and facilities. This prioritization of school subjects may then reflect the lack of resources allocated to the PA sessions, including financial resources, facilities, and staff expertise.

We had assumed that schools with written policies on creating a stimulating school environment and new opportunities to be physically active would ensure that they invested in the implementation of obligatory PA. We found this significant association in the 2018 survey, where written guidelines were important both for the increased use of qualified staff in the sessions and for an increased number of competence-enhancing measures linked to the PA scheme. We found no such associations in the 2014 survey. This may be due to the fact that the scheme was not as well established at the time and that by 2018, a larger proportion of schools had conducted competence-enhancing measures. This assumption aligns with implementation and development of practices based on political guidelines being dynamic processes that contain different phases, and the effects of these are seen when work is undertaken purposefully over a long period of time [34].

The results may indicate that schools’ written guidelines are important in order to conduct competence development and also to use this competence in the sessions. Written guidelines can therefore be a crucial measure to raise competence levels in relation to PA sessions [11,35]. In a study by Haug et al. [36], based on data from 68 schools and 1347 pupils who participated in the Health Behaviour in School-Aged Children (HBSC) survey in 2005/2006, the results showed that schools that had written guidelines for physical activity had a higher proportion of pupils who were active during break times than schools that did not have written guidelines. In line with McLeroy and colleagues’ (1988) socioecological model, the findings show that a national state policy with increased time for physical activity in school can affect the individual pupil’s level of physical activity through allocating time for compulsory physical activity during school hours. The findings also support the importance of the intermediate processes in the socioecological model in that written guidelines for physical activity at institutional level (school) seem to have resulted in a higher-quality effort through using trained staff that are skilled in ensuring involvement and interaction between all participating pupils and thereby possibly stimulating activities that are perceived as more fun and manageable for the participants.

This study has some methodological considerations. One limitation is that the data are based on self-reported assessments from head teachers, thus suspect to recall bias and social desirability responses, and may depend on their knowledge of the integration of the PA scheme across grades. Despite the fact that we do not know how detailed their knowledge is, previous Norwegian studies have shown that self-reported data from school leadership on organizational and structural facilitation can predict pupils’ participation in activities during school recess [35,36,37]. The survey was sent out to 138 schools in 2014 and 136 schools in 2018, with a wide geographical spread and with a response rate of 49%. We do not have the specific reasons why the schools did not answer the questions directed at the schools’ leadership (school-level survey), but general reasons given for refraining from participating in the study (both on a school level and pupil level) are linked to the school taking part in too many studies, and this stole time and resources [38]. Thus, we do not know if there is a selection bias related to interest in school health-promotion actions in the sample. Even though the sample size is too limited to be able to generalize the findings, the data are still extensive enough to contribute more knowledge on how Norwegian schools manage the PA scheme and the differences and challenges that exist.

## 5. Conclusions

The introduction of the obligatory PA scheme was an important political commitment to a more active everyday life for children and young people in Norway [1]. From an international perspective, adding 76 h of compulsory physical activity to the overall school day of Norwegian 5th–7th graders in addition to compulsory PE-classes is an important step from a policy level to increase the opportunities for physical activity in school. It aligns with the Global Action Plan on Physical Activity 2018–2030, which emphasizes the need to strengthen the provision of good-quality physical education and more positive experiences and opportunities for active recreation, sports and play, applying the principles of the whole-of-school approach in primary educational institutions [9]. Considerable resources are used each year in implementing the PA scheme in the 13 years the scheme has existed. Still, it has been the subject of very little research and evaluation [16], and there are so far no published scientific studies on the topic. This study, carried out five years and nine years after the introduction of the scheme, thus contributes valuable information about how the scheme is practiced in Norwegian schools. A lack of investment in skills development, large student groups, and little use of staff with subject-specific competence indicate that Norwegian schools do not have sufficient resources to ensure that the aims and the minimum requirements of the PA scheme are fulfilled. The two measurement years seem to follow many of the same patterns. The main difference is that in 2018, we see even less use of qualified staff. This is cause for concern in terms of the quality of the sessions, as research suggests that who is responsible and the organization of the sessions are key factors for the quality, activity level and health benefits [26]. Overall, the findings indicate that there is a need for a more comprehensive evaluation of the PA scheme to assess the extent to which the scheme contributes to a more varied and active everyday school life.

## Figures and Tables

**Figure 1 ijerph-19-14312-f001:**
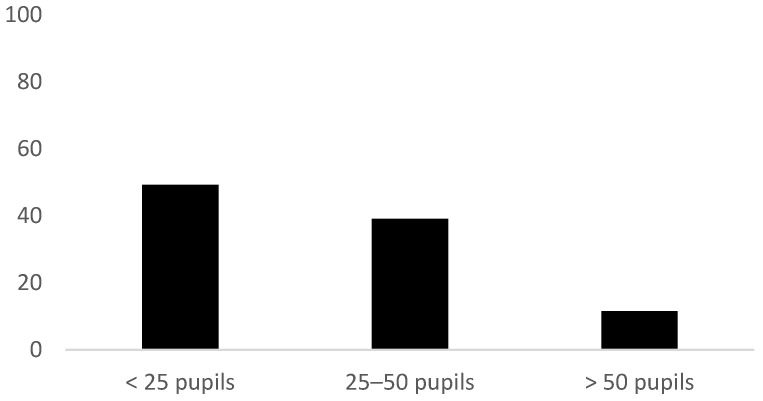
Group size in PA classes (*n* = 69). Data from 2014.

**Figure 2 ijerph-19-14312-f002:**
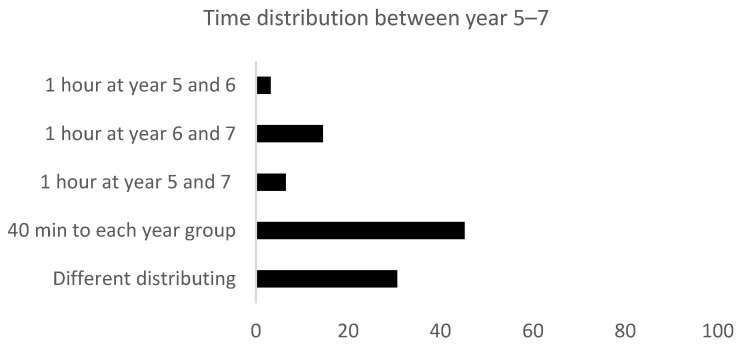
An overview of how the schools distributed the allocated 76 h of obligatory PA time in years 5–7. The columns in the figure show distribution between schools (%) in relation to the various alternatives for organizing allocated hours (*n* = 69) Data from 2014.

**Figure 3 ijerph-19-14312-f003:**
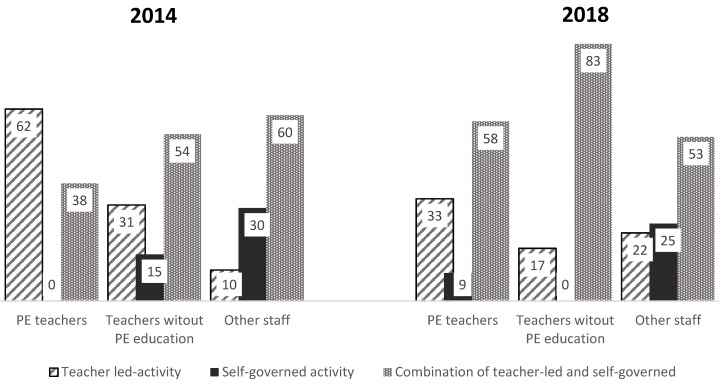
An overview of the kind of staff and the means of organizing (percentage distribution) in PA classes in 2014 (*n* = 69) and 2018 (*n* = 65). Significant differences between how different staff organized the lessons in 2014 (*p* = 0.001) were observed.

**Figure 4 ijerph-19-14312-f004:**
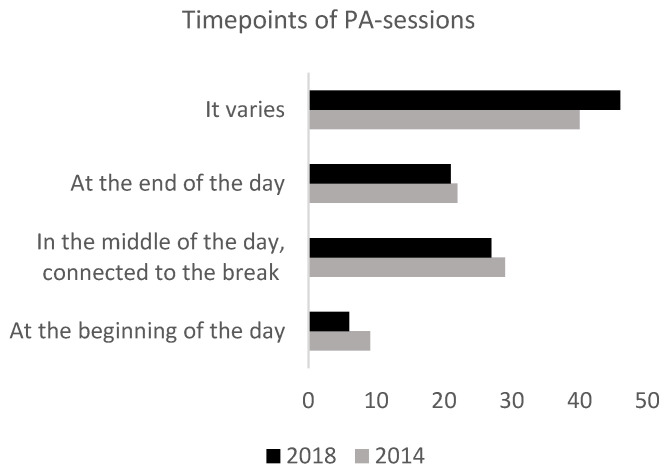
An overview of what time of day the PA sessions were conducted in 2014 and 2018. The figure shows the percentage of responses from the schools (2018: *n* = 65, 2014: *n* = 69).

**Table 1 ijerph-19-14312-t001:** Overview of who was responsible for conducting the PA sessions in 2014 and 2018.

Survey Year	Pedagogical Staff Who Teach PE (%)	Pedagogical Staff Who Do Not Teach PE (%)	Other Staff (%)	*n*
2014	51	19	30	68
2018	31	19	50	65

## Data Availability

The University of Bergen is the data bank manager for the HBSC study. Please contact Ellen Haug for data requests.

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
