# Peer review of "An Investigation of the Implementation of Obligatory Physical Activity Classes for 5th–7th Grade in Norway"

_ijerph, 2022, doi:10.3390/ijerph192114312_

Round 1

Reviewer 1 Report

This article is well written and well structured, and presents an interesting implementation by the Norwegian government to fight the obesity and the "being hooked to the screens" by the younger ones.

The authors presented a study based on two surveys that were carried out in 2014 and 2018 on PA, but unfortunately these same surveys had only 49% of responses, which itself shows some fragility in the results presented in this article.

However, apart from a few confusing sentences, the article offers a unique reading on a very important topic.

Overall, after some minor corrections, it is an interesting article.

Errors/questions:

Line 24 - The sentence is somewhat confusing, please review it.

Line 46 - The sentence is somewhat confusing, please review it.

Line 76 - The sentence is somewhat confusing, please review it.

Line 140: "The purpose of this study is to explore how a larger sample of Norwegian ..." - Here the right word would perhaps be  analyze rather than explore.

Line 145 - There must be an introductory paragraph explaining what will be listed next in the chapter/section.

Line 267 - The sentence is somewhat confusing, please review it.

Line 322: "We are not aware of other countries that on a national level have integrated mandatory physical activity classes in addition to compulsory PE classes for pupils at this age." - The authors say that "they don't know" is something very negative, I believe that some research was done to verify if there is or has been something similar in other countries right?

Line 333: "and... and..." - Please review this.

Line 355: "Our results showed that in 2018, only 30% of the schools used staff with formal PE competence for the PA-scheme and, in addition, half the schools reported that they use staff without any form of pedagogical competence for these sessions." - And was it possible to verify if there is an improvement in these 30% of students? I can't see this information in the document.

Line 416: "...but many of the same patterns are repeated between 2014 and 2018..." - Which for example? it would be interesting to see here some listed/detailed

Line 476: "From an international perspective, adding 76 hours of compulsory physical activity to the overall school day of Norwegian 5 – 7 graders in addition to compulsory PE classes, is a bold step from a policy level to increase the opportunities for physical activity in school." - Why is it a bold step in an international level? How do you support this claim?

Line 486: "This study, carried out five years and nine years after the introduction of the scheme, thus contributes valuable information about how the scheme is practised in Norwegian schools." - But does it correspond to the objective proposed in line 140? if yes, it should be more detailed here in my opinion.

Line 491: "Several of the minimum requirements can be difficult to be met when the responsibility lies with unskilled staff." - Which?

Reviewer 2 Report

Did you detect inappropriate self-citations by authors?

There are instances of self-referenced material, but not inappropriate in this instance. A good paper I'd be confident to proceed with.

Round 2

Reviewer 3 Report

I thank the authors for their revision in-line with reviewer requests. It is much improved.